# Routine childhood immunization coverage, timeliness, dropout, and missed opportunities in Northern Ghana: Evidence from a community-based survey in Tamale Metropolis, Ghana

**Matthew Konlan**[1]*, **Chrysantus Kubio**[2☯], **Sheila Agyeiwaa Owusu**[3,4☯],
**Fuseini Mahama**[2‡], **Benedict Ofori Appiah**[2‡], **Michael Rockson Adjei**[5☯],
**Oheneba Boadum**[6☯], **Martin Nyaaba Adokiya**[7☯], **Peter Gyamfi Kwarteng**[8☯],
**Maxwell Oduro Yeboah**[9☯], **Hilarius Asiwome Kosi Abiwu**[10]

**1** Division of Global Health, School of Population and Global Health, The University of Western Australia, Perth, Western Australia, **2** Northern Regional Health Directorate, Ghana Health Service, Tamale, Ghana, **3** Department of Paediatrics and Child Health, School of Medicine, University for Development Studies, Tamale, Ghana, **4** Clinical Research Department, London School of Hygiene and Tropical Medicine, United Kingdom, **5** WHO Country Office for Ghana, Accra, Ghana, **6** Department of Clinical Anatomy, Sam Houston State University, College of Osteopathic Medicine, Conroe, Texas, United States of America, **7** Department of Epidemiology, Biostatistics and Disease Control, School of Public Health, University for Development Studies, Tamale, Ghana, **8** Department of Health and Nutrition, UNICEF Ghana, Tamale, Ghana, **9** Department of Nursing, Northern Regional Hospital, Tamale, Ghana, **10** National Malaria Elimination Programme, Public Health Division, Ghana Health Service, Accra, Ghana

☯ These authors contributed equally to this work
‡ These authors also contributed equally to this work
* matthew.konlan@research.uwa.edu.au

## Abstract

Routine childhood immunization plays a crucial role in reducing vaccine-preventable diseases and child mortality. However, sustained progress requires not only high coverage but also timely administration, continuity across multi-dose schedules, and minimization of missed opportunities for vaccination. Persistent subnational inequities, post-COVID-19 disruptions, and gaps in service delivery threaten progress toward Immunization Agenda 2030 targets. This study assessed routine immunization coverage, timeliness, dropout rates, and missed opportunities for vaccination among children aged 12–59 months in the Tamale Metropolis of northern Ghana. We conducted a descriptive analysis of 1,020 children aged 12–59 months, drawn from a larger community-based cross-sectional study involving 1,577 children aged 0–59 months in the Tamale Metropolis. Multi-stage sampling was used to select mothers or caregivers of eligible children in this study. Trained research assistants collected data from November to December 2022 through face-to-face interviews, and a review of home-based records using a validated questionnaire. Findings were summarized using descriptive statistics. Our findings showed that 35.0% of the children were fully vaccinated according to the complete national immunization schedule, while 10.0% received no basic vaccines. Crude coverage for vaccines administered at 6, 10,

**Data availability statement:** All relevant data are within the paper and its Supporting Information files.

**Funding:** The authors received no specific funding for this work.

**Competing interests:** The authors have declared that no competing interests exist.

and 14 weeks exceeded 70%., Notably, coverage was low for MenA (44.3%), MR2 (44.1%), and Rota3 (43.2%). A significant proportion of children received vaccine doses either before the recommended age, at intervals shorter than recommended, or delayed by more than two months. Dropout was high and varied across subdistricts, with Rota 1–3 reaching 51.6%. Among children with missed opportunities for vaccination, about 50% of missed doses were corrected within one to two months. Persistent gaps in timeliness, high dropout rates, and missed opportunities for vaccination threaten Ghana's ability to achieve IA2030 targets. Strengthening digital immunization registries, defaulter tracing systems, targeted interventions, and enhanced engagement with healthcare providers is essential to ensure comprehensive vaccine coverage.

## Introduction

Routine childhood immunization is one of the most effective public health interventions for reducing morbidity and mortality from vaccine-preventable diseases (VPDs) [1]. Globally, immunization prevents an estimated 4.4 million deaths each year [2]. Since the inception of the Expanded Program on Immunization (EPI) in 1974, global efforts have saved an estimated 154 million lives and contributed to a 40% reduction in global infant mortality [3]. Despite these gains, low- and middle-income countries (LMICs), including Ghana, face persistent challenges in achieving and sustaining optimal immunization coverage.

In recent years, following the COVID-19 pandemic, LMICs have focused on restoring immunization coverage to pre-COVID-19 pandemic levels [4,5]. However, the number of zero-dose children (i.e., children who have not received any routine vaccinations) has continued to rise, alongside under-served and missed communities [4,6,7]. In response, the Global Immunization Agenda 2030 (IA2030) prioritizes equity, data-driven action, and reaching populations that are consistently left behind, particularly zero dose and under-vaccinated children [8]. Achieving these goals requires timely, localized and context-specific evidence, as national-level averages often obscure substantial subnational and community-level inequities [9,10]. Community-level assessments are therefore critical to identifying gaps in immunization service delivery, including delayed vaccination, dropout from multi-dose schedules, and missed opportunities during health facility encounters.

Ghana adopted the EPI strategy in 1978 [11], resulting in substantial improvements in immunization coverage—from 19% in 1988 [12] to 79% in 2008 [13]. However, these gains plateaued and subsequently declined over the past decade (2014–2024). National coverage fell from 77% in 2014 [14] to 73% in 2022 [15]. While the COVID-19 pandemic likely contributed to this decline, evidence suggests that immunization coverage in Ghana was already suboptimal prior to the pandemic due to structural and systemic challenges, including health system fragility, population mobility, and financing constraints [16]. Recent outbreaks of VPDs in Ghana [17,18] suggest that current immunization efforts are insufficient to meet the IA2030 goals [19], underscoring the need for renewed and targeted strategies to strengthen routine immunization systems [20].



To reverse declining trends and improve immunization coverage in Ghana, effective strategies would have to be grounded in up-to-date, high-quality and context-specific evidence on vaccine uptake, particularly in low-performing regions, including the Northern Region [15]. Although routine administrative data systems provide readily available data for immunization monitoring and decision-making, these systems are often affected by data quality limitations in LMICs, including inaccuracies and incomplete reporting [21]. Consequently, household and community-based surveys play a critical role in validating administrative data and, in some contexts, provide more reliable estimates of true immunization coverage and service performance [22].

In addition, the Demographic and Health Survey (DHS), and Multiple Indicator Cluster Survey (MICS) also provide robust national and regional immunization estimates for Ghana, they are not designed for district, sub-district or Community-based Health Planning and Services (CHPS) level decision-making. In Northern Ghana, where routine reports suggest high coverage, the 2022 GDHS nevertheless indicates comparatively elevated zero-dose prevalence (9.0%) [15], and recent measles outbreaks in the Northern Region (including Tamale) [18] point to localized immunization gaps. In line with IA2030's call for subnational equity monitoring and the urban immunization evidence base and as part of a post-COVID-19 recovery effort, we conducted a community-based assessment in the Tamale Metropolis to generate district-, subdistrict- and CHPS-level estimates of coverage, timeliness, dropout, and missed opportunities among children aged 12–59 months to inform strategic planning.

## Materials and methods

### Study design and population

We conducted a descriptive cross-sectional study of 1,020 children aged 12–59 months to enable us to estimate full immunization coverage, describe immunization patterns, and pinpoint operational bottlenecks in routine childhood immunization in the Tamale Metropolis from November to December 2022. This study was carved out of a larger community-based cross-sectional study involving 1,577 children aged 0–59 months in the Tamale Metropolis. Children less than 12 months were excluded because they had not reached the age at which completion of the routine immunization schedule could be assessed.

### Study area, sampling and data collection

This study was conducted in the Tamale Metropolis of the Northern Region, Ghana. We estimated an initial sample of 1,512, using the WHO survey sample size calculator version 2 [23]. The calculation was based on the following assumptions: 95% coverage for all basic immunizations (Northern Regional target), a precision of ±5% (95% confidence interval), a design effect (DEFF) of 2, a minimum of 5 children per cluster, and an intra-cluster correlation coefficient (ICC) of 1/3. We further assumed that 10% of eligible respondents will be unavailable or will decline the survey and that an average of four (4) households would need to be visited to identify one eligible child.

During survey implementation, we successfully enrolled 1,577 eligible children. The overall sample was allocated to each selected community in a cluster using probability proportional to size, based on the estimated number of eligible children per community provided by the Tamale Metropolis Health Directorate.

For the present analysis, we restricted the sample to 1,020 children aged 12–59 months. This age group was selected in accordance with the Ghana national immunization schedule [24] and WHO recommendations [25], as it allows valid estimation of full immunization coverage. In effect, including children aged 0–11 months was likely to systematically underestimate full immunization coverage because children in that age group had not yet completed the recommended vaccination schedule.

Survey participants were selected using a four-stage sampling design. At stage one, all the 63 clusters were purposively selected from the four sub-Metros (Tamale Central, Bilpeila, Nyohini, and Vittin) in the Tamale Metropolis. Clusters comprised of demarcated CHPS zones in the Tamale Metropolis, regardless of whether they had a physical structure or not. A complete list of these zones and their communities was obtained from the Tamale Metro Health Directorate.



At stage two, a list of all communities within each cluster was compiled and coded. One community per cluster (63 in total) was then selected using simple random sampling.

At stage three, the starting point for household selection within each sampled community was determined by selecting a random direction from the community centre (using the pen-spinning method). Houses along this direction were counted to the community boundary, and the first household was chosen at random. Subsequent households were selected using a systematic interval (every 3rd house in urban communities and every 2nd house in peri-urban communities) to account for differences in housing density until the required number of households was reached.

At stage four, where a selected house contained multiple households with eligible children, simple random sampling was used to select one household. If more than one eligible child resided in the selected household, one child was randomly selected for inclusion. A total of 24 households per cluster were sampled.

Data were collected by trained research assistants through face-to-face interviews with caregivers and review of home-based vaccination records. Questionnaire development and survey implementation were guided by the updated WHO vaccination coverage cluster survey reference manual (2018) [26], which provides standardized questions, indicators, and analytic guidance. A detailed description of the study area, and data collection has been published from the same survey [27]. The study was reported in accordance with the STROBE guidelines for cross-sectional studies [28].

## Indicator definitions and reporting recommendations

### Indicator definitions

1. We defined Missed Opportunities for Vaccination (MOV) due to non-simultaneous vaccination as documented vaccination visits in which a child received one or more vaccines but not all the vaccines for which they were eligible [29]. For example, in a visit where Penta3 is given but OPV3 and PCV3 are not, even though these doses were not completed, and the minimum interval had passed since their last dose. This is an approach endorsed by the WHO [30].

2. Dropout was defined as the proportion of children aged 12–59 months who received a dose in a multi-dose sequence or a preceding vaccine in the schedule but failed to receive a subsequent or the final dose in the sequence [31].

3. We validated vaccine doses using exact dates on the home-based immunization records. For each antigen, we compared the date on the home-based record with the minimum age and interval defined by the Ghana EPI schedule. Any vaccination given before the minimum age or before the minimum interval between doses within a series had elapsed was considered early and classified as invalid [32,33].

4. Crude coverage referred to the proportion of children who are vaccinated against one or more specified VPDs at the date before the survey [32].

5. Valid coverage referred to the proportion of children aged 12–59 months whose vaccination by the doses followed the earliest recommended age and minimum interval between doses are counted in the numerator [32].

6. Full immunization is commonly used as a better measure of the full benefit of immunization compared to DTP3, particularly in LMICs [34]. Previously in the early years of the WHO EPI, only four basic vaccines (BCG, DTP (3 doses), OPV (3 doses) and measles-containing vaccine) against six diseases were included in national immunization schedules of LMCIs. For a child to be fully vaccinated, they needed to have received all these doses. However, the number of vaccines included in national immunization schedules now varies significantly. As such a fully vaccinated child can refer to children who have received all four basic vaccines, but by replacing DTP with the pentavalent vaccine, or can stringently be defined as children who have received all the vaccines in the country's immunization schedule in the time period of the participants of the study [29]. In this study, based on the Ghana national immunization schedule, we restricted fully vaccinated children to those who had received one dose of BCG, three doses each of Penta, OPV,



Rota, and PCV, one dose of IPV and a dose of MR1. This provides a more comprehensive estimate of fully vaccinated children in the study setting but may not be comparable with many global reports.

7. Timeliness of vaccine doses was defined as too early (given <28 days interval), timely (28 days), <2 months late, 2+months late and timing unknown using dates when the ensuing vaccine doses were received or given.

## Reporting recommendations

The WHO recommends that routine immunization coverage be tabulated according to the source of information [32] and coverage should be classified as crude and valid [25]. In this study, although we adhered strictly to these recommendations, in interpreting the results, crude estimates are used to enable us understand immunization service availability, contact with the health system and the immunization program reach, regardless of timing of vaccine doses.

## Statistical analysis

Data were analysed using the WHO Vaccination Coverage Quality Indicators (VCQI) tool [35], and Stata 15.1 statistical software (StataCorp) [36]. The VCQI is a flexible, freely available Stata package recommended by the WHO for estimating immunization coverage from surveys—estimates generated by VCQI account for the complex survey design. The WHO issued a survey white paper on immunization, which describes how to handle the following issues during data cleaning and indicator development: definitions of eligible population and denominators for each indicator; handling evidence from tick marks; handling imperfect date values; handling missing values or 'unsure' or 'do not know'; steps to differentiate routine immunization (RI) doses from supplementary immunization activity (SIA) doses; definitions of valid doses, calculation of confidence intervals and how many decimal places to report [32]. We aligned our analysis with the prescriptions in this white paper. The denominator for each antigen included all age-eligible children. As a result, denominators vary by vaccine, as coverage was calculated based on the number of children old enough to be eligible for that specific dose at the time of the survey. Vaccination cards that were 'not seen' were not excluded; instead, all doses were coded as missing for those children. For children with cards but missing dates for specific vaccines, those doses were classified as missing. Only card-documented dates were used to determine valid coverage and timeliness. In our analysis and reporting of the results, we weighted the outcomes involving all respondents. Given the potential for weighting to under- or over-estimate coverage in sub-analyses because the sample is no longer representative of the population, we did not weight samples when the unit of analysis was a subset of the total sample.

Missed opportunities for vaccination were calculated using the number of eligible facility visits as the denominator, defined as all documented contacts during which the child was age-eligible for at least one due vaccine. The numerator comprised visits where the eligible vaccine(s) were not administered, regardless of later correction. Initially, we focused on vaccines requiring scheduling visits. As such, we did not initially collect data on the vaccine doses administered at birth. However, pictures of the biodata and vaccination pages for children with home-based records were taken during data collection. In our quest to ensure that coverage estimates reflect the situation in Tamale and based on the high card retention rate, we reviewed the pictures and entered the relevant data for birth doses. Thus, our analysis and reporting of birth doses are based on children with eligible biodata and vaccination pages. Although the malaria vaccine is currently part of the routine immunization schedule in Ghana (**Table 1**), it was excluded from our analysis and reporting. At the time of this study, the malaria vaccine had not been introduced in the Northern Region.

In this study, our final analysis produced coverages based on card and history. However, the history coverages for all the vaccine doses were zero, given the high card retention rate. Therefore, we report coverage based on the card only in this study.

## Ethical considerations

This study was approved by the Navrongo Health Research Centre Institutional Review Board (NHRC-IRB) on 16th December 2022 (Approval ID: NHRCIRB495). Following submission of the initial application, comments from the IRB were

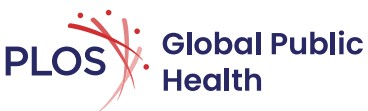

**Table 1. Routine immunization schedule in Ghana.**

| Vaccine | Year introduced | Schedule | | | | | | | |
|---|---|---|---|---|---|---|---|---|---|
| | | At birth | 6 weeks | 10 weeks | 14 weeks | 6 months | 7 months | 9 months | 18 months |
| BCG | 1978 | 1 | | | | | | | |
| OPV | 1978 | 0 | 1 | 2 | 3 | | | | |
| Penta (DPT, Hib, Hep) | 2002 | | 1 | 2 | 3 | | | | |
| PCV | 2012 | | 1 | 2 | 3 | | | | |
| Rota | 2012 | | 1 | 2 | 3 | | | | |
| IPV | 2018 | | | | 1 | | 2 | | |
| MR | 2013 | | | | | | | 1 | 2 |
| YF | 1992 | | | | | | | 1 | |
| MenA | 2016 | | | | | | | | 1 |
| Malaria | 2019 | | | | | 1 | 2 | 3 | 4 |

*Note*: BCG, bacillus Calmette–Guérin; OPV, oral polio vaccine; Penta, pentavalent vaccine (includes diphtheria, pertussis, tetanus, Haemophilus influenzae type b, and hepatitis B vaccines); PCV, pneumococcal conjugate vaccine; IPV, Inactivated Polio Vaccine; MR, Measles-Rubella; YF, Yellow Fever, menA, meningococcal serotype A.

addressed and a revised application was submitted. Data collection commenced after the revisions had been completed but prior to receipt of the formal approval letter. The IRB subsequently granted approval for the study, covering all data collected. Verbal informed consent was obtained from all caregivers prior to data collection. The consent process involved reading a standardized participant information sheet approved by the NHRC-IRB, explaining the study purpose, procedures, voluntary participation, confidentiality, and the right to withdraw at any time. Verbal informed consent was approved by the IRB and used in place of written consent because the study posed minimal risk, involved no collection of personal identifiers, and was conducted in settings where varying literacy levels could make written consent burdensome or inadvertently coercive. To document consent, data collectors recorded each participant's verbal agreement directly on the Samsung Mobile tablets used to collect the data. In a few instances where required, a field supervisor served as a witness to the verbal consent process.

## Results survey sample description

Of the 1,577 caregivers of children aged 0–59 months, 1,020 caregivers of children aged 12–59 months were included in this analysis, representing 64.7% (1,020/1,577) of the total sample. Of this number, 94.0% (959/1,020) were female, and the same proportion were the biological mothers of the children whose immunization status was evaluated (94.0%; 959/1,020). Nearly all the mothers (99.9%; 1,019/1,020) attended ANC during pregnancy. About 89.9% (917/1,020) delivered their children in a health facility, and slightly over half (52.5%; 535/1.020) have not received any formal education. A little less than three-quarters (70.9%; 723/1,020) were employed. Slightly over half were aged 25–24 years (55.3%; 564/1,020). About 92.5% (944/1,020) of the children received their most recent vaccination at a static point in a health facility. A detailed summary of the survey sample is shown in **Table 2**.

## Routine immunization coverage in children aged 12–59 months

**Table 3** describes both crude and valid immunization coverage by vaccination card for children aged 12–59 months in the Tamale Metropolis. Crude coverage was high for most vaccines, ranging from 89.7% (915/1,020; 95% CI: 85.0, 93.0) for OPV1 to 72.6% (740/1,020; 95% CI: 67.4, 77.4) for yellow fever vaccine. Coverage for OPV0 was comparatively lower at 54.2% (553/1,020; 95% CI: 48.6, 57.8). Remarkably, coverage was low for MenA (44.3% (452/1,020); 95% CI: 39.1, 49.7)

**Table 2. Health-related information and socio-demographic characteristics of survey participants.**

| Variables | % (n = 1,020) |
|---|---|
| **Attended ANC (yes)** | 99.9 |
| **Place of birth** | |
| Home/traditional birth attendant | 7.5 |
| Public health facility | 89.9 |
| Private health facility | 2.6 |
| **Respondent's relationship to child** | |
| Father | 4.5 |
| Mother | 94.0 |
| Other (sister, grandmother, stepmother) | 1.5 |
| **Respondent's sex (female)** | 94.0 |
| **Respondent's education** | |
| None | 52.5 |
| Primary | 20.5 |
| Secondary | 20.5 |
| Tertiary | 6.6 |
| **Employment (unemployed)** | 70.9 |
| **Respondent's age group (years)** | |
| <=24 | 19.2 |
| 25-34 | 55.3 |
| 35-44 | 23.5 |
| 45-54 | 1.9 |
| >=55 | 0.1 |
| **Where child received most recent vaccination** | |
| Static point in a health facility | 92.5 |
| Outreach point in neighbourhood | 6.5 |
| Other (during campaign) | 1.0 |

and MR2 (44.1% (450/1,020); 95% CI: 38.7, 49.6). For the multi-dose vaccines, coverage waned with age. For example, coverage decreased from 89.6% (914/1,020; 95% CI: 85.0–92.9) for Penta 1 to 84.3% (860/1,020; 95% CI: 79.5–88.1) for Penta3. Only 35.0% (357/1,020; 95% CI: 29.1–41.5) were fully immunized with all the basic vaccines, while 10.0% (102/1,020; 95% CI: 6.7–14.6) of the children had received none of the basic vaccines.

## Timeliness of vaccinations

**Fig 1** shows the timeliness of vaccinations without point estimates. Among children whose vaccination cards were seen, a significant majority had received vaccine doses at the recommended age or intervals between doses. However, a concerning proportion of them received some vaccine doses before the recommended age or with too short intervals between doses. This was common for Rota1, PCV1, OPV1 and Penta1. A substantial proportion received vaccine doses 2 + months late, particularly for MR1, YF, IPV, PCV3, OPV3 and Penta3.

## Dropout rate

As shown in **Fig 2**, the overall dropout rate was high in the Tamale Metropolis, ranging from 51.6% (471/912) for Rota 1 to Rota 3 to 5.9% (54/914) for Penta1 to Penta 3. Dropout varied significantly by subdistrict. Bilpeila recorded the highest

Table 3. Weighted crude and valid immunization coverage among children aged 12-59 months (N = 1,020) in the Tamale Metropolis, Ghana, 2022.

| Vaccine | Crude coverage (%) and 95% CI | | Valid coverage (%) and 95% CI | |
|---|---|---|---|---|
| BCG | 83.0 | (78.5, 86.8) | 80.0 | (75.2, 84.0) |
| OPV0 | 53.2 | (48.6, 57.8) | 43.1 | (38.9, 47.5) |
| Penta1 | 89.6 | (85.0, 92.9) | 88.3 | (83.8, 91.7) |
| OPV1 | 89.7 | (85.0, 93.0) | 88.0 | (83.4, 91.5) |
| PCV1 | 89.5 | (84.8, 92.9) | 87.5 | (83.0, 91.0) |
| Rota1 | 89.4 | (84.7, 92.8) | 86.9 | (82.4, 90.4) |
| Penta2 | 88.6 | (84.1, 92.0) | 85.2 | (80.5, 88.9) |
| OPV2 | 88.2 | (83.7, 91.6) | 84.4 | (79.8, 88.2) |
| PCV2 | 87.5 | (82.9, 91.1) | 82.7 | (78.0, 86.6) |
| Rota2 | 87.5 | (83.0, 91.0) | 80.2 | (75.5, 84.2) |
| Penta3 | 84.3 | (79.5, 88.1) | 72.1 | (67.1, 76.5) |
| OPV3 | 82.0 | (77.0, 86.0) | 68.7 | (63.5, 73.5) |
| PCV3 | 82.6 | (77.9, 86.6) | 70.5 | (65.2, 75.3) |
| Rota3 | 43.2 | (36.7, 50.1) | 36.2 | (30.6, 42.1) |
| IPV | 81.4 | (76.1, 85.7) | 79.3 | (74.1, 83.7) |
| MR1 | 73.4 | (68.0, 78.2) | 63.8 | (58.6, 68.6) |
| YF | 72.6 | (67.4, 77.4) | 68.9 | (63.6, 73.8) |
| MenA | 44.3 | (39.1, 49.7) | 39.7 | (34.9, 44.8) |
| MR2 | 44.1 | (38.7, 49.6) | 40.6 | (35.6, 45.9) |
| Fully vaccinated | 35.0 | (29.1, 41.5) | 25.2 | (20.8, 30.2) |
| Not vaccinated (zero-dose) | 10.0 | (6.7, 14.6) | 11.1 | (7.1, 15.7) |
| Weighted sample size | 1,020 | | 1,020 | |
| CI = Confidence Intervals | | | | |

dropout for Penta1 to Penta3 (7.5%; 18/240) and PCV1 to PCV3 (10.4%; 25/240). Tamale Central recorded the highest dropout for OPV1 to OPV3 (11.1%; 27/244). Vittin recorded the highest dropout for Rota1 to Rota3 (74.8%; 172/230).

## Missed opportunities for simultaneous vaccination

Considering visits that a child was eligible for any dose of the vaccines, missed opportunities for simultaneous vaccination varied across sub-districts. Nyohini recorded the highest proportion of no missed opportunities for vaccination (19.2%; 38/198), as well as the highest proportion of fully corrected MOSVs (36.9%; 73/198) compared with the other sub-districts. In contrast, Vittin had the highest proportions of partially corrected MOSVs (47.4%; 110/232) and uncorrected missed opportunities for vaccination (32.3%; 75/232). Across all the subdistricts, Rota 3 had the highest percentage of uncorrected missed opportunities for vaccination (Fig 3).

Fig 4 illustrates the cumulative distribution of the time to correction in days, for missed opportunities for vaccination by dose and subdistrict. The red lines represent the 50th percentile, illustrating that in some cells, 50% of the corrections occurred within one or two months of the missed opportunity. These were common for BCG and IPV vaccines.

## Discussion

Immunization coverage is a crucial metric for assessing and monitoring the performance of EPI programs and the overall health system. This study provides updated community-level estimates of routine childhood immunization coverage, timeliness, dropout rates and missed opportunities for vaccination among children aged 12–59 months in the Tamale

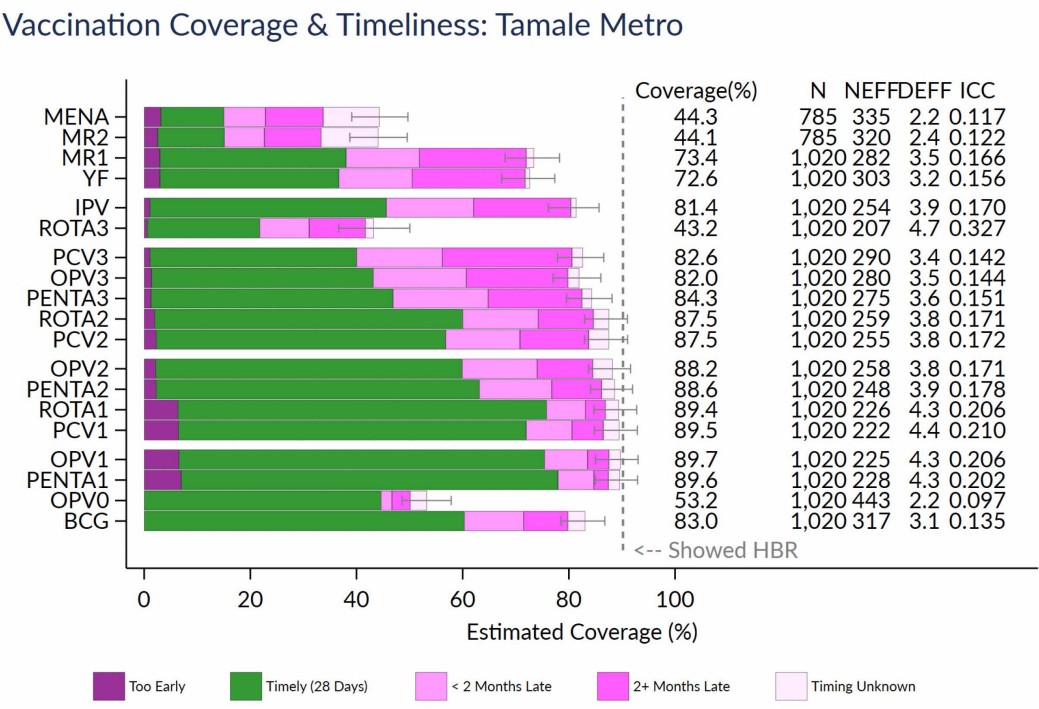

Abbreviations: HBR: Home-based record  NEFF: Effective sample size  DEFF:

**Fig 1.  Timeliness of vaccination for children aged 12-59 months with a date of vaccination in the Tamale Metropolis.**

Metropolis. The findings reveal substantial variation across vaccines and stages of the immunization schedule, with important implications for program planning, targeted interventions, and monitoring progress toward Immunization Agenda 2030 targets.

Reporting both crude and valid coverage provides complementary insights into immunization system performance. Crude coverage reflects overall reach, whereas valid coverage, based on WHO minimum age and interval criteria, captures whether children received immunologically effective doses. Given that mistimed doses do not confer optimal protection, valid coverage is the more appropriate indicator for assessing true programme performance and identifying quality gaps. However, in this study we focus on crude coverage as our purpose was to understand the overall immunization performance in the Tamale Metropolis.

Coverage varied substantially by vaccine and age at administration. Crude coverage rates for vaccines administered at 6, 10, and 14 weeks were relatively high but remained slightly below national and regional targets (coverage of at least 95%) [37], and estimates of the 2022 Ghana Demographic and Health Survey [15]. These patterns suggest successful early engagement with immunization services, likely reflecting the accessibility of infant health services and routine attendance at child welfare clinics [38]. However, coverage declined slightly for vaccines scheduled later in the immunization calendar, suggesting that there are inherent challenges in maintaining consistent vaccination uptake as children age. A pronounced drop was observed for subsequent doses within multi-dose vaccine series, particularly for the third dose of the rotavirus vaccine. The marked discrepancy between coverage for the third rotavirus dose and other third-dose vaccines administered at the same visit warrants further investigation to identify the underlying causes of this gap. Notably, completion of the rotavirus series dropped sharply, with Rota3 coverage substantially lower than earlier doses. Similar

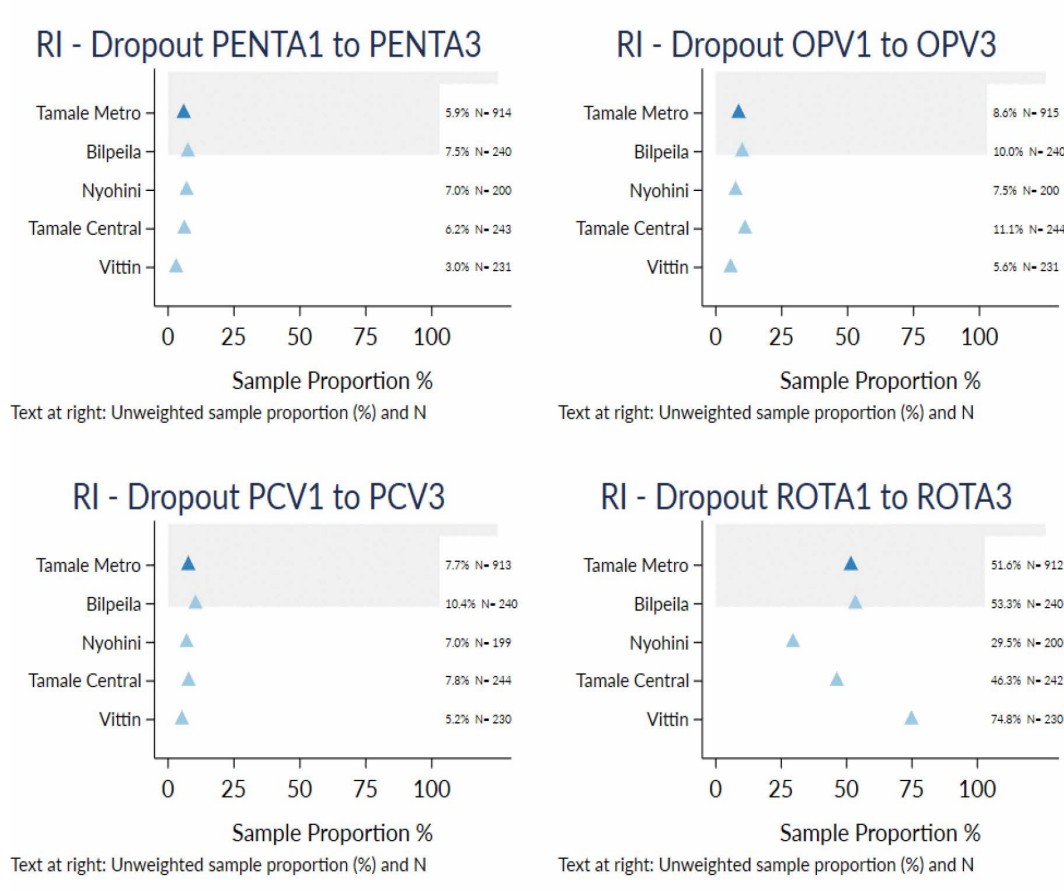

**Fig 2. Percentage dropout between first and third doses for selected multi-dose vaccines among children aged 12-59 months in the Tamale Metropolis, Ghana, 2022.**

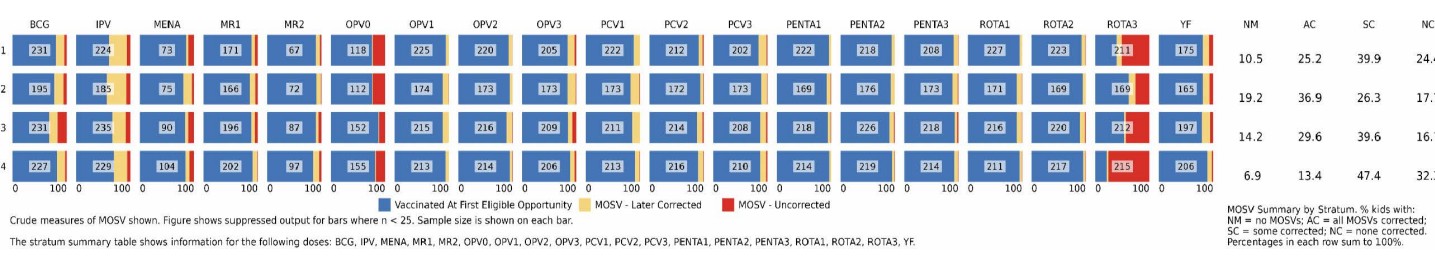

**Fig 3. Percentage of children vaccinated at the first eligible opportunity, and percent who experienced one or more missed opportunities, whether later corrected.** The numbers in the centre of each cell portray the number of children in that subdistrict (row) who had 1+ vaccination visits when age eligible to receive that dose. 1 = Bilpeila, 2 = Nyohini, 3 = Tamale Central, 4 = Vittin.

trends have been reported in other studies, where initial uptake is high but declines progressively across later doses [39–41]. Coverage for vaccines given at 18 months (MenA and MR2) was notably low (below 50%). These low coverages signal a risk for the rebirth of VPD outbreaks [22], as a significant proportion of the children are likely to be unprotected. Both demand- and supply-side strategies need to be explored to increase coverage for vaccines given later in the

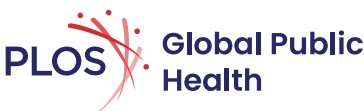

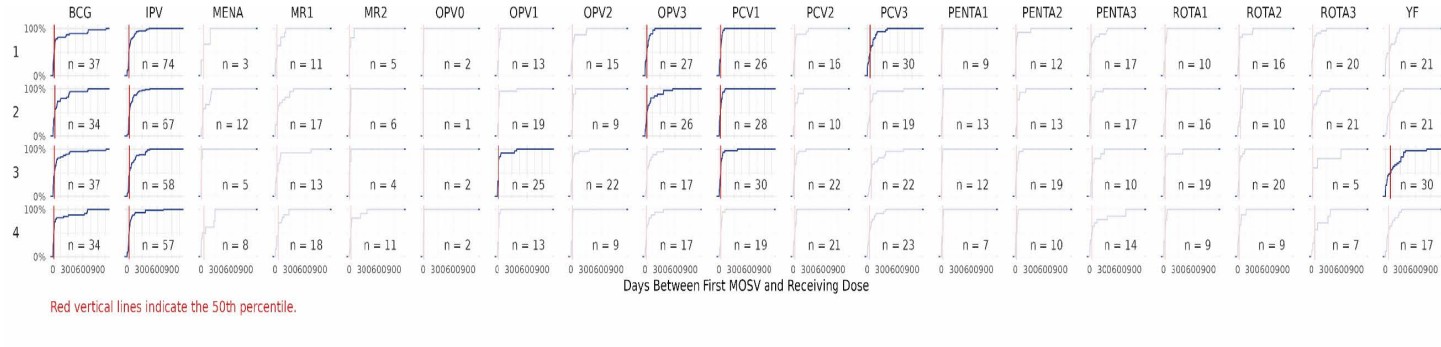

Red vertical lines indicate the 50th percentile.

Crude measures of MOSV shown. Figure shows suppressed output for bars where n < 25.

**Fig 4. Cumulative distribution of the time to correction in days for missed opportunities for vaccination by dose and subdistrict.** 1 = Bilpeila, 2 = Nyohini, 3 = Tamale Central, 4 = Vittin.

schedule, as these may be due to caregiver complacency, access challenges, and caregiver forgetfulness or the feeling that immunization has been completed [42].

A notable proportion of children aged 12–59 months received one or more vaccine doses either before the recommended minimum age or with intervals shorter than recommended, resulting in invalid doses, while others experienced significant delays in vaccination. The occurrence of invalid doses may indicate underlying service delivery quality gaps, including insufficient adherence to minimum age and interval requirements, provider workload pressures, or inconsistent supervision and quality control mechanisms within immunization sessions [43], which has implications for attaining IA2030 strategic priority one of immunization for primary health care. These gaps can undermine the accuracy and reliability of vaccine administration, ultimately compromising the effectiveness of the immunization programme and the protective immunity achieved in children [44].

Early dosing was most frequently observed for vaccines administered at initial infant contacts, including OPV1, PENTA1, PCV1, and ROTA1, whereas delayed vaccination was more common for vaccines scheduled later in the immunization calendar, particularly MR1, YF, IPV, and third doses of multi-dose vaccines such as OPV3, PCV3, and PENTA3. These patterns indicate inconsistent adherence to recommended vaccination schedules across antigens and stages of the immunization schedule. Administering vaccines too early or out of sequence can lead to suboptimal immune responses, potentially leaving children inadequately protected against VPDs while delayed vaccination prolongs periods during which children remain susceptible to vaccine-preventable diseases [25].

Variation in vaccination timeliness across subdistricts further points to the influence of local service delivery practices, provider adherence to schedules, caregiver awareness, and resource availability. Improving timeliness will, therefore, require strengthening provider training and supportive supervision to reinforce age-appropriate vaccination practices and correct interval spacing [45]. In addition, caregiver-focused interventions, including appointment reminders and education on the importance of adhering to scheduled visits, may help reduce both early and delayed dosing [46]. Finally, robust routine monitoring systems capable of identifying early, delayed, and missed doses are essential for improving vaccination quality and timeliness and for supporting targeted corrective actions at the facility and subdistrict levels [47]. Adherence to recommended vaccination schedules is essential to ensure optimal protection against VPDs [48].

Dropout between doses in multi-dose vaccine series was substantial and varied by vaccine type and sub-district. The highest dropout was observed for the rotavirus vaccine, with more than half of children failing to complete the Rota1 to Rota3 series (51.6%). In contrast, dropout for other multi-dose vaccines administered at similar visits, such as pentavalent and oral polio vaccines, was considerably lower. These findings indicate important gaps in follow-up and continuity of care after initial vaccine receipt, particularly for vaccines requiring completion within a narrow time window [42]. Marked

sub-district variation in dropout rates further highlights geographic inequities in immunization performance within the Tamale Metropolis. Notably, Vittin recorded rotavirus dropout rates nearly twice those in other sub-districts, highlighting localized weaknesses in service delivery or follow-up mechanisms. Such disparities suggest that dropout is influenced not only by caregiver behaviour but also by local health system factors, including accessibility of services, effectiveness of defaulter tracing, and consistency of vaccine availability. Although the present study did not analytically examine individual-level determinants, previous research in Ghana has shown that incomplete vaccination is associated with caregiver characteristics, including marital status, religion, and gender of the child, which can influence care-seeking behaviours and limit the possibility of dropout [39,49,50]. For instance, children of married caregivers, female children, and those from Christian households have shown lower dropout risks. Additionally, children with immunization cards are less likely to drop out of the immunization programme in Ghana [51]. These factors may influence caregivers' ability to adhere to scheduled visits and maintain continuity within the immunization programme.

The magnitude of rotavirus vaccine dropout observed in this study substantially exceeds estimates reported elsewhere in sub-Saharan Africa. A multi-country analysis reported an average rotavirus dropout rate of 10.8%, with estimates ranging from 2.8% in Rwanda to 37.7% in Uganda [52]. The markedly higher dropout observed in the Tamale Metropolis suggests context-specific challenges related to health system organization, service delivery practices, and population dynamics. Given that rotavirus vaccination requires timely completion early in infancy, missed or delayed follow-up visits may disproportionately affect series completion. Further facility-based and mixed-methods studies are needed to better understand the relative contributions of service-level factors, caregiver engagement, and documentation practices to the high dropout observed, and to inform targeted strategies to improve completion of multi-dose vaccine schedules.

Missed opportunities for simultaneous vaccination varied substantially across subdistricts, underscoring important within-district variation in immunization service delivery. Nyohini recorded the highest proportion of missed opportunities for vaccination but also the highest proportion of missed opportunities for vaccination that were fully corrected. This pattern suggests that, although missed opportunities were frequent in this subdistrict, follow-up mechanisms such as caregiver return visits, outreach activities, or defaulter tracing, were relatively effective in facilitating timely correction of missed doses. In contrast, Vittin exhibited the highest proportions of partially corrected and uncorrected missed opportunities for vaccination, pointing to persistent gaps in follow-up and continuity of care after an initial missed opportunity.

Analysis of the cumulative distribution of time to correction provided additional insight into health system responsiveness following missed opportunities. Across several dose–subdistrict combinations, 50% of missed opportunities were corrected within one to two months, indicating that many caregivers returned to health services within a relatively short period after a missed vaccination. This pattern was particularly evident for BCG and IPV. For BCG, early correction is consistent with Ghana's immunization practice of administering the BCG vaccine during early infancy, often at postnatal or child welfare clinics, and likely reflects strong early-life contact with health services. Similarly, relatively rapid correction for IPV may be explained by its administration alongside other injectable vaccines, increasing the likelihood of correction at subsequent routine visits.

This study highlights strengths and gaps in routine childhood immunization delivery in Tamale that are relevant to Ghana's EPI performance reviews and IA2030 monitoring. High coverage for early vaccines suggests successful initial contact with immunization services. However, the marginal declines in coverage for later doses, high dropout rates and missed opportunities for vaccination, points to weaknesses in continuity of care. There is the need for EPI monitoring frameworks to place greater emphasis on second year-of-life vaccination, dropout tracking, timeliness, and MoV indicators, rather than focusing predominantly on early-dose coverage. Sub-district variation in performance further underscores the value of decentralized, data-driven reviews that enable targeted corrective actions in underperforming areas.

This study has several strengths. The use of a large, community-based sample enabled detailed assessment of routine childhood immunization coverage, timeliness, dropout rates, and missed opportunities for vaccination across multiple

sub-districts within the Tamale Metropolis. This sub-district–level granularity provides a clearer understanding of immunization performance that is not attainable through routine administrative data alone and strengthens the relevance of the findings for local EPI planning and monitoring. Nonetheless, several limitations should be considered when interpreting the findings. First, the analysis relied on home-based vaccination records, which minimises recall bias but may limit representativeness for cardless children without vaccination cards. As a result, missed opportunities for vaccination and coverage may be underestimated among children without cards, and incomplete or missing records could have introduced measurement error in estimates of coverage, timeliness, and missed opportunities. Second, the study primarily quantifies coverages, timeliness, dropout rates and missed opportunities for vaccination without deeply exploring the underlying causes or associations, such as socio-economic barriers, health system inefficiencies, or individual caregiver beliefs and knowledge. Consequently, causal inferences regarding factors driving these patterns cannot be made. Third, although estimates included birth doses, they were based on children whose biodata pages and vaccination record pictures were taken during data capture. Other estimates were based on card records. Reliance on card data may underestimate the true immunization coverage in the Tamale Metropolis. Finally, given that this study used a cross-sectional design, we captured data at a single point in time, limiting our ability to establish causal relationships. Despite these limitations, the study provides robust, community-level evidence on immunization coverage, continuity, timeliness, and service delivery gaps in a rapidly urbanizing setting, offering valuable insights to inform targeted interventions and strengthen routine immunization systems.

## Conclusions

The study identifies significant challenges in vaccination coverage among children aged 12–59 months in the Tamale Metropolis, characterised by low coverage for vaccines given at 18 months and high dropout rates and missed opportunities for vaccination, particularly for the rotavirus vaccine. Despite high initial vaccine uptake, sustaining consistent immunization schedules remains a challenge. These gaps highlight the need for improved tracking, healthcare provider training, and community engagement to ensure that all children receive the full benefits of immunization programs. Addressing these challenges and strengthening tracking systems and community engagement will be essential to achieve the targets of the IA2030.

## Supporting information

**S1 Data. Database of manuscript: Immunization coverage, timeliness, dropout and missed opportunities for vaccination among children aged 12–59 months in the Tamale Metropolis, Ghana.**
(XLSX)

**S1 File. Validated questionnaire.**
(DOCX)

## Acknowledgments

We thank our respondents for their time and the field officers for their service during fieldwork.

## Author contributions

**Conceptualization:** Matthew Konlan, Hilarius Asiwome Kosi Abiwu.

**Data curation:** Matthew Konlan, Hilarius Asiwome Kosi Abiwu.

**Formal analysis:** Matthew Konlan.

**Investigation:** Matthew Konlan.

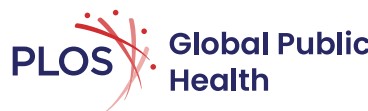

**Methodology:** Matthew Konlan, Chrysantus Kubio, Fuseini Mahama, Benedict Ofori Appiah, Michael Rockson Adjei, Oheneba Boadum, Martin Nyaaba Adokiya, Peter Gyamfi Kwarteng, Maxwell Oduro Yeboah, Hilarius Asiwome Kosi Abiwu, Sheila Agyeiwaa Owusu.

**Project administration:** Matthew Konlan.

**Software:** Matthew Konlan.

**Supervision:** Matthew Konlan, Fuseini Mahama, Benedict Ofori Appiah, Hilarius Asiwome Kosi Abiwu.

**Writing – original draft:** Matthew Konlan.

**Writing – review & editing:** Matthew Konlan, Chrysantus Kubio, Fuseini Mahama, Benedict Ofori Appiah, Michael Rockson Adjei, Oheneba Boadum, Martin Nyaaba Adokiya, Peter Gyamfi Kwarteng, Maxwell Oduro Yeboah, Hilarius Asiwome Kosi Abiwu, Sheila Agyeiwaa Owusu.

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
