## [Decision Letter · Decision Letter 0]

17 Nov 2025

PGPH-D-25-02884

Immunisation Coverage, Dropout Rate, Timeliness and Missed Opportunities for Vaccination Among Children Aged 12-23 Months in Tamale Metropolis, Ghana

Dear Dr. Konlan,

Thank you for submitting your manuscript to PLOS Global Public Health. After careful consideration, we feel that it has merit but does not fully meet PLOS Global Public Health’s publication criteria as it currently stands. Therefore, we invite you to submit a revised version of the manuscript that addresses the points raised during the review process.

I was fortunate to get 5 reviewers - you'll note some similarities and differences across what they've mentioned. I think a quick look over grammar, spelling, spacing, and consistency in acronyms is useful, but you do not need to spend a lot of space in your response detailing that.

Additionally, I understand that there are 5 reviewers, and there is some overlap in what they mention. Feel free to respond with something like "we've already addressed this point in response to another reviewer (see response above to reviewer 3)".

I would also spend more time in your response with a bit more details on your sample, than on adding much more to your discussion.

We look forward to receiving your revised manuscript.

Kind regards,

Abram L. Wagner, PhD, MPH

Academic Editor

Journal Requirements:

1. In the ethics statement in the Methods, you have specified that verbal consent was obtained. Please provide additional details regarding how this consent was documented and witnessed, and state whether this was approved by the IRB

2. Please include a complete copy of PLOS’ questionnaire on inclusivity in global research in your revised manuscript. Our policy for research in this area aims to improve transparency in the reporting of research performed outside of researchers’ own country or community. The policy applies to researchers who have travelled to a different country to conduct research, research with Indigenous populations or their lands, and research on cultural artefacts. The questionnaire can also be requested at the journal’s discretion for any other submissions, even if these conditions are not met.  Please find more information on the policy and a link to download a blank copy of the questionnaire here: https://journals.plos.org/globalpublichealth/s/best-practices-in-research-reporting. Please upload a completed version of your questionnaire as Supporting Information when you resubmit your manuscript.

3. Please send a completed 'Competing Interests' statement, including any COIs declared by your co-authors. If you have no competing interests to declare, please state "The authors have declared that no competing interests exist". Otherwise please declare all competing interests beginning with the statement "I have read the journal's policy and the authors of this manuscript have the following competing interests:"

4. Please amend your detailed Financial Disclosure statement. This is published with the article. It must therefore be completed in full sentences and contain the exact wording you wish to be published.

5. In the online submission form, you indicated that Data for this manuscript is readily available and can be obtained from the corresponding author upon reasonable request.

3. Uploaded as supplementary information.

6. Please provide separate figure files in .tif or .eps format.

7. Please remove any file of the article processing charge (APC) from your File Inventory which is visible to our Editors. Any questions about the APC or billing process should be directed to authorbilling@plos.org to ensure they are kept separate from the editorial decision-making process.

Reviewers' comments:

Reviewer's Responses to Questions

**Comments to the Author**

1. Does this manuscript meet PLOS Global Public Health’s publication criteria? Is the manuscript technically sound, and do the data support the conclusions? The manuscript must describe methodologically and ethically rigorous research with conclusions that are appropriately drawn based on the data presented.? Is the manuscript technically sound, and do the data support the conclusions? The manuscript must describe methodologically and ethically rigorous research with conclusions that are appropriately drawn based on the data presented.

Reviewer #1: Yes

Reviewer #2: Yes

Reviewer #3: Partly

Reviewer #4: Partly

Reviewer #5: Yes

2. Has the statistical analysis been performed appropriately and rigorously?

Reviewer #1: Yes

Reviewer #2: No

Reviewer #3: Yes

Reviewer #4: No

Reviewer #5: Yes

3. Have the authors made all data underlying the findings in their manuscript fully available (please refer to the Data Availability Statement at the start of the manuscript PDF file)?

The PLOS Data policy requires authors to make all data underlying the findings described in their manuscript fully available without restriction, with rare exception. The data should be provided as part of the manuscript or its supporting information, or deposited to a public repository. For example, in addition to summary statistics, the data points behind means, medians and variance measures should be available. If there are restrictions on publicly sharing data—e.g. participant privacy or use of data from a third party—those must be specified.requires authors to make all data underlying the findings described in their manuscript fully available without restriction, with rare exception. The data should be provided as part of the manuscript or its supporting information, or deposited to a public repository. For example, in addition to summary statistics, the data points behind means, medians and variance measures should be available. If there are restrictions on publicly sharing data—e.g. participant privacy or use of data from a third party—those must be specified.

Reviewer #1: Yes

Reviewer #2: No

Reviewer #3: No

Reviewer #4: Yes

Reviewer #5: Yes

4. Is the manuscript presented in an intelligible fashion and written in standard English?

Reviewer #1: Yes

Reviewer #2: Yes

Reviewer #3: Yes

Reviewer #4: Yes

Reviewer #5: Yes

Reviewer #1: GENERAL COMMENTS:

This manuscript describes a study to assess routine immunization coverage, dropout rates, timeliness, and missed opportunities for vaccination among children aged 12 to 23 months in the Tamale Metropolis in Ghana.

It is a relevant study because it provides the rationale for countries to routinely perform these analyses and it describes some of the available tools for conducting this type of analysis, specifically the "WHO Vaccination Coverage Cluster Survey Reference Manual" and the "Vaccination Coverage Quality Indicators" tool.

However, I fell that the paper does not provide a sufficient discussion of what are the remedial steps that a country should take when there are deficiencies and weaknesses in routine immunization coverage. There is a limited discussion of remedial actions to address timeliness of vaccination in Lines 288 to 299, and a discussion of the possible role of caregiver characteristics that could lead to increased dropout rates (Lines 307-313. However, the paper would be far more useful if the authors could expand on these sections and particularly on the section dealing with missed opportunities for vaccination (Lines 334-340) Can the authors provide any concrete description of what occurred in Gambia as a response to this study?

SPECIFIC COMMENTS:

Line 1: Capitalize the "t" in "Timeliness"

Line 52 Place "(MoV)" after "Missed opportunities for vaccinations..." This allows you to then use the abbreviation, "MoV" as an abbreviation later in the text.

Line 57 Change to "...Penta1 to Penta3."

Reviewer #2: Thank you for submitting your manuscript, “Immunisation Coverage, Dropout Rate, Timeliness and Missed Opportunities for Vaccination Among Children Aged 12–23 Months in Tamale Metropolis, Ghana,” to PLOS Global Public Health. This is an important and well-executed study addressing a critical area of child health in a low-resource context. The topic is highly relevant to the journal’s mission to advance equitable global health through evidence-based research.

Below are detailed comments intended to help strengthen the manuscript before publication.

Overall Assessment

The manuscript presents a methodologically sound and ethically rigorous cross-sectional study examining immunisation coverage, dropout rates, and missed opportunities in northern Ghana. The research question is clearly articulated, the sampling and data collection methods are appropriate, and the analysis is competently executed using WHO-recommended tools (VCQI and Stata).

The findings provide valuable local evidence to inform immunisation system strengthening under the Immunisation Agenda 2030 (IA2030) and contribute to the broader global discourse on vaccine equity in low- and middle-income countries.

Overall, the methodology is sound and ethically conducted, but the analytical scope is narrow, offering limited insights into causal or contextual determinants. The paper is well written, clearly structured, and largely meets PLOS Global Public Health’s publication standards. Revisions focused on analytical depth, data transparency, and stylistic refinement will enhance its quality and impact.

Major Comments

Analytical depth and statistical rigour

- The analysis is appropriate for a descriptive cross-sectional study. However, it would be strengthened by including basic inferential analyses (e.g., bivariate or multivariable logistic regression) to explore factors associated with dropout or missed opportunities for vaccination.

- If inferential analysis is not feasible, please explicitly justify this choice and emphasise that the study’s intent is descriptive and diagnostic.

- Clarify whether survey weights and cluster design effects were accounted for in the statistical analysis. This is particularly relevant given the multi-stage sampling design.

- The manuscript does not report any sensitivity or robustness checks, which would strengthen confidence in the estimates given the reliance on vaccination cards and potential missing data.

Data availability and transparency

- The current Data Availability Statement (“available upon reasonable request”) does not fully meet PLOS’s open data policy. Please deposit the anonymised dataset in a public repository or provide a valid justification for restricted access (e.g., ethical or legal constraints).

- Update the statement accordingly with a persistent link or DOI.

Ethical and Reporting Standards

- Ethical approval was obtained (Navrongo Health Research Centre IRB), informed consent was secured, and data availability is declared.

- Please include the date of ethical approval.

- Please explain why verbal consent rather than written consent was used.

Discussion and implications

- Expand the discussion to include policy and programmatic implications. For instance, how could your findings inform Ghana’s EPI performance reviews or IA2030 monitoring frameworks?

- Reflect briefly on health system and social determinants that may underlie dropout and missed opportunities (e.g., access barriers, service quality, gender dynamics).

Conclusion

- The conclusion somewhat restates findings rather than extrapolating actionable recommendations.

Limitations section

- The limitations are clearly acknowledged, but you could strengthen this section by discussing potential biases (recall bias, card completeness) and the cross-sectional design’s inability to infer causality.

Minor Comments

Language and style:

- The manuscript is written in clear, standard English, but a light copyedit is recommended to correct minor grammatical, punctuation, and stylistic inconsistencies (e.g., “optimum” → “optimal,” consistent vaccine name formatting, and spacing before citations).

Presentation:

- Ensure consistent use of terms such as dropout rate, missed opportunities, and coverage across tables and text.

- Some sentences in the Discussion are repetitive and could be streamlined for conciseness.

References:

- Ensure consistent formatting (some entries mix sentence-case and title-case).

- Double-check URLs (e.g., WHO links) for functionality.

- Reference [37] author name should be capitalised (“Singh, J.” not “singh, j.”).

Areas for improvement (minor language and stylistic edits):

- While the manuscript is generally polished, there are a few minor grammatical and stylistic issues that should be corrected at revision for clarity and precision:

-- Article usage and syntax:

Example: “Coverage must be optimum to realise its life-saving potential” → should read “Coverage must be optimal to realise its life-saving potential.”

Example: “Invalid doses were common, reaching 2.6% (29/1,111) for OPV” → could be rephrased for smoother readability (“Invalid doses were relatively common, accounting for 2.6% of OPV administrations”).

- Consistency in capitalization and terminology:

--Ensure consistent capitalization of vaccine names (e.g., “Penta3 to MR1” vs “penta3 to MR1”) and sub-districts.

--Harmonise spelling (e.g., “organisation” vs “organization”) according to PLOS’ preference for US English.

- Minor typographical or punctuation issues:

-- A few misplaced commas and extra spaces (e.g., “immunisation coverage, dropout rate and missed opportunities for vaccination among children aged 12-23 months in the Tamale Metropolis” — add commas for readability).

-- Occasional missing spaces before citations (e.g., “[12]and further to 73%”).

- Sentence tightening for conciseness:

--Some sentences are overly long or repetitive in the Discussion section; these could be broken into shorter, more direct statements for improved readability.

Summary Recommendation

This is a valuable and timely study that provides evidence to support improvements in immunisation coverage and service delivery in Ghana. With minor revisions to enhance analytical clarity, data transparency, and language polish, the manuscript will be well suited for publication in PLOS Global Public Health.

Line-by-Line (or Section-by-Section) Comments

Title and Abstract

Title: Consider capitalising consistently (e.g., Timeliness rather than timeliness).

Abstract, lines 30–36: “Coverage must be optimum to realise its life-saving potential” → “Coverage must be optimal to realise its life-saving potential.”

Abstract, Results: Add a short quantitative summary of overall fully vaccinated percentage for clarity (e.g., “Only 27.7% of children were fully vaccinated”).

Abstract, Conclusion: The phrase “pose significant challenges in the study setting” could be streamlined: “pose major challenges to achieving full immunisation coverage.”

Introduction

Line 69: “Routine childhood immunisations have significantly reduced morbidity and mortality associated with vaccine-preventable diseases (VPDs) globally.” → You could add a recent 2024 WHO/UNICEF global estimate to strengthen this opening claim.

Lines 74–80: Long sentence beginning “While LMICs are currently working…” could be split for clarity.

Line 88: “Conflicts, fragility and inadequate funding” → consider serial comma: “conflicts, fragility, and inadequate funding.”

Line 95: “Effective strategies have to be devised, based on current rates of vaccine uptake…” → could be simplified: “Effective strategies must be informed by up-to-date evidence on vaccine uptake.”

Methods

Lines 111–114: Clarify whether sampling accounted for cluster design effects in analysis (use of survey weights).

Lines 123–124: “A detailed description of the study area, sampling and data collection have been captured in a publication from the same survey [22].” → should be singular: “has been captured.”

Lines 126–146 (Definitions): Excellent clarity — but please ensure definitions follow WHO phrasing exactly, e.g., “valid dose” and “invalid dose.”

Lines 149–159: The phrase “Data were analysed With the VCQI tool” — lowercase “with.”

Line 177 (Ethics): Good inclusion. Suggest adding the date of ethical approval if available.

Results

Line 195: “Immunisation coverage among children aged 12–23 months…” – consider adding N=1,111 to reinforce sample context.

Table 2: Excellent detail, but column headings could be reformatted for readability (e.g., “Valid Coverage (%)” and “95% CI”). Ensure consistent decimal precision (one decimal place).

Lines 214–219 (Timeliness): Sentence “Vittin recorded the highest proportion of invalid doses…” could be simplified for flow: “The Vittin sub-district recorded the highest proportion of invalid doses (5.1%), while Bilpeila recorded the lowest (0.9%).”

Fig. 1: Ensure consistent capitalization of vaccine series names (Penta1–3, MR1).

Discussion

Lines 263–266: “Crude coverage rates… were high. Though slightly below national and regional targets…” → Combine into one smoother sentence: “Crude coverage rates were high but remained slightly below national and regional targets of 95%.”

Lines 283–289 (Timeliness): “A notable proportion of the children… received invalid doses.” → Add a short comment linking to potential service delivery quality issues.

Lines 300–312: The discussion of dropout determinants could cite Ghana-specific socio-demographic studies or GHS reports beyond [36] to enrich interpretation.

Line 324: Typo: “where children where children were eligible” → remove repetition (“where children were eligible”).

Lines 335–338: “Health workers must be equipped…” – this paragraph is strong; consider tightening the last sentence for impact: “Reliable vaccine supply chains and supportive supervision are essential to prevent stockouts and ensure timely vaccination.”

Lines 341–354 (Strengths & limitations): Well written; you could add one line acknowledging that the exclusion of birth doses and reliance on card data may underestimate true coverage.

Conclusion

Lines 356–365: The conclusion could end more assertively by linking to policy or IA2030 targets, e.g., “Strengthening tracking systems and community engagement will be essential to achieve Ghana’s IA2030 coverage targets.”

Reviewer #3: This is strong, methodologically sound, and valuable study that provides granular, actionable data on immunization program performance in the Tamale Metropolis, Ghana. The research moves beyond simple coverage estimates to diagnose key programmatic failures: high dropout rates, missed opportunities for vaccination (MoV), and issues with timeliness (invalid doses), closely linked to clinical effectiveness and safety! The focus on these specific, measurable gaps with regional variations is a significant strength and provides a clear roadmap for targeted, context-specific interventions. While this paper is of high quality and suitable for publication in PLOS Global Public Health, it requires Major Revisions before it can be accepted. 1. he current Data Availability Statement ("Data... can be obtained from the corresponding author upon reasonable request") is not compliant with PLOS policy. The authors must deposit the minimal, anonymized dataset required to replicate the study's findings in a public repository (e.g., Dryad, Zenodo, Figshare) and update the Data Availability Statement in the manuscript to provide the repository name and persistent DOI/accession number; 2. As this is a cross-sectional observational study, it should be reported in accordance with the STROBE guidelines. Please add a sentence to the Methods section confirming adherence to STROBE and please also provide a completed STROBE checklist as a supplementary information file. 3. The authors state a "validated questionnaire" was used. To ensure full reproducibility, which is a core tenet of PLOS, please provide this questionnaire as a supplementary information file. Minor issues: 1. Definition of fully vaccinated is fantastic but results in a low 16.0% coverage that might be miscompared to the traditional (and less stringent) global benchmark (e.g., DTP3/MCV1). Please add a sentence in the Methods or Discussion to briefly contextualize this, noting that your definition is based on the full national schedule and is therefore more comprehensive than many standard global reports, therefore not comparable. 2. Clarify denominators in table 2. In Table 2, the denominators for coverage correctly decrease for later vaccines (e.g., n=790 for Penta1, n=207 for MR2). This is because not all children in the 12–23-month cohort were old enough for the 18-month vaccines. Please add one sentence in the "Statistical analysis" or Results section (before Table 2) to explain this for the reader (e.g., "Denominators vary by vaccine, as coverage was calculated based on the number of children old enough to be eligible for that specific dose at the time of the survey. 3. our results show a striking drop-off for Rota3 (43.6% coverage) and a massive 52.9% MoV rate. This is a much poorer performance than Penta3 (81.9%) or PCV3 (74.0%), which are given at the same visit. This is a critical quality-of-care finding. Please add 1-2 sentences to the Discussion pointing out about the need to further explore the root-causes on this specific discrepancy (e.g., Rota-specific stockouts, confusion over oral vs. injectable administration, etc.). 5. The methods refer to a previous publication "[22]" for a detailed sampling description. While this is fine, the manuscript would be rigorous if you add 2-3 sentences to the "Study area, sampling and data collection" section summarizing the multi-stage sampling process. Typos and formatting should also be fixed on line 344, Table 4 and Figure 1

Reviewer #4: This research paper sought to provide updated evidence on immunisation coverage, drop-out rates and missed opportunities in the Tamale Metropolis of Ghana, using a descriptive analytical approach. While the paper has good potential, the methodological and analytical uncertainties require major revision. I have provided additional information below for the authors' reference for future submissions.

Introduction

The authors provide a good overview of immunisation coverage and the need for updated evidence. While contextual Ghanaian emphasis is well laid out, authors do not provide information on why this research matters for the Tamale Metropolis. This is important because reports from Ghana’s DHS already provide information on immunisation trends, from which dropouts and missed opportunities can be inferred. If this research focuses on Tamale, why is this the case? Are there peculiar coverage issues in Ghana’s Northern region that warrant this additional research and new data collection? Is there consistently low vaccination uptake in the sub-context? Are current nationally representative data sources inadequate? Is there a gradual increase in certain childhood illnesses that warrant advocacy for better immunisation efforts in the metropolis?

Lines 101-103: It is unclear what authors mean by “ household surveys are limited, particularly in northern Ghana, where disparities in vaccine uptake persist”. Ghana’s demographic and health surveys (DHS) as well as multiple indicator cluster surveys (MICS) are household surveys which cover the Northern region as well. The statement is therefore not substantive. Is there literature to substantiate the claim? Furthermore, it is important to provide additional information on why a new community-based cross-sectional survey is warranted when information can be derived from the DHS/MICS. Are there key peculiarities for the Tamale Metropolis that needs to be highlighted?

Lines 76 and 79: Do you mean missed "opportunities" rather than "communities"?

Lines 89: Can you provide the full meaning of VPDs since it appears for the first time in the work?

Materials and methods

This section overlooks many essential details. First, the section labelled "study design" is misleading, given that it captures information mainly on sample size. Even for this, there is no substantive explanation for the sample used for this research. How did the authors determine the sample size of 1602 for their survey? What contributed to the final subsample of 1111 children? The attached data has rows up to 1534. Why is this the case?

Second, little information is given on how the multi-stage sampling was done. How many stages? How was the sampling frame derived? What approach was used in identifying and selecting participants? Were participants sampled randomly? Additionally, the timeframe for data collection is unclear and somewhat suspicious. The authors report that they collected the data between 21st December 2022 and 10th January 2023; however, the attached data (Excel) shows dates before this reported period (29th November - 12th December). Is there a reason for this? The authors also report that the cross-sectional cluster survey consisted of 1602 children aged 0-59 years but it is concerning that out of this sample, 1111 were children aged 12-23 months (about 70% of the whole sample). Was the sampling approach not random? There is also no justification for focusing on those aged 12-23 months.

Third, the terms need to be well-defined, and authors should provide references to support them. For instance, missed opportunities are defined as children not receiving all age-appropriate vaccines per the immunisation schedule. Do these include drop-outs from multi-dose vaccines or when the child misses out on all the doses of a vaccine entirely? Or does this include missing the age-appropriate timing for the vaccine? Do authors capture children who are aged 12-17 months but have not yet taken MR2 (taken at 18 months) as missed opportunities? This is not well defined. Additionally, how do authors deal with this (children aged 12-17 months who have not taken MR2 yet) when calculating "full vaccination"? Are they automatically captured as not fully vaccinated? For invalid doses, the term only captures instances where the dose is given before the right age or when its <28days in between doses. Why are those given after the scheduled dates/time not included? How did authors validate data on invalid doses? This particular term (invalid doses) is used consistently to infer timeliness of vaccine uptake in the results section, however, the concept of vaccine timeliness extend beyond what "invalid doses (as currently defined)" capture.

Fourth, the authors do not provide adequate information on how the data was weighted. Why were subsets not weighted? The authors also indicate that they did not include vaccines taken at birth. However, these vaccines are included in internationally defined vaccination markers. Why were those vaccines (e.g. BCG) excluded? To what extent does the omission of BCG vaccines and OPV0 vaccines affect vaccine coverage estimates?

Lastly, the ethical considerations need further clarification. Authors need to provide additional information on what they mean by the rights of respondents. How was voluntary participation, anonymity and confidentiality ensured? The data attached has information on respondents' telephone numbers and house addresses, which is not appropriate given the importance of anonymity in research. Did participants also consent to having their personal details publicly available? How was data managed and protected?

Results

Lines 185-186: The interpretation does not come out clearly. Do you mean that 99% of the mothers attended antenatal care? Authors could take time to break long sentences into simple sentences for better clarity.

The results in Table 2, specifically those for "fully vaccinated" and "not fully vaccinated," require revision. Authors stated in the methods section that “not fully vaccinated” is the opposite of being “fully vaccinated”, which has a binary inference unless otherwise. The estimates for "not fully vaccinated" do not match up then. If about 27.7% of children have been fully vaccinated, then 72% have not been fully vaccinated. However, the percentages reported do not support this observation. Does the authors’ definition of “not fully vaccinated” mean zero doses (if this is the case, the estimates do not measure up either when more than 80% of children have taken at least one vaccine)? Authors need to provide more precise definitions and re-analyse the work to ensure appropriate estimates. Furthermore, in Table 2, the weighted sample size is 111 for both crude coverage and valid coverage. Is this a mistake?

The concept of vaccine timeliness used in this research does not align with international usage of the term and may require the use of an alternative term that fits the study – “invalid doses”. I observe that the percentages reported as invalid doses in Table 3 do not substantiate the estimates reported as "Valid coverage" in Table 2, especially when valid coverage is calculated as “Crude coverage – invalid doses”. For example, the average 2.5% observations reported as invalid for Penta vaccines in Table 3 seem under-estimated. From Table 2, Penta 3 crude coverage is 81.9% and the valid coverage is 63.8%. Given how authors define valid coverage, this means that invalid dose for Penta 3 alone (difference between coverage and valid coverage) will be about 18%. This estimate is higher than what authors report as the average invalid dose for Penta vaccines. It could be that authors are not providing better explanation for their estimates. Authors are encouraged to provide additional supplementary information to explain how they arrived at the estimates.

Drop-out rates: Are drop-out rates based on crude or valid coverage? Authors seem to provide overreported estimates. For instance, the descriptive estimates from Table 2 show that uptake of OPV1 is 91.3% and OPV3 is 72.1% (based on crude coverage). This means that the drop-out rate is ~19.2% but the authors report 22.3% (the estimates do not tally either when valid coverage is used). The same issue is observed for the other multi-dose vaccines. It is also unclear why authors would want to draw estimates between Penta 3 and MR1 as these are entirely different vaccines. Furthermore, there is no information for other multi-dose vaccines such as Rota and PCV. Why is this the case?

Missed opportunities: Authors could report percentages of actual missed opportunities. If the missed opportunities were corrected (which is unclear what this means), then they are no longer missed opportunities.

Since the authors collected information on some demographic characteristics, why was this not captured in the analysis?

Discussion

The discussion is good, however, it could be strengthened further. The authors could explain why reporting crude and valid vaccination coverage matters and recommend which is more appropriate. Given the issues raised in the methods and results section, the authors may need to revisit the discussion write-up.

Reviewer #5: Review report

Title: Immunisation Coverage, Dropout Rate, Timeliness and Missed Opportunities for Vaccination Among Children Aged 12–23 Months in Tamale Metropolis, Ghana

Suggested title: Routine Childhood Immunisation Coverage, Dropout, and Missed Opportunities in Northern Ghana: Evidence from a Community-Based Survey in Tamale Metropolis.

General Assessment: The authors present a cross-sectional survey on immunisation performance among children in parts of northern Ghana. The study attempts to provide recent, community-level data on vaccination coverage, timeliness, dropout rates, and missed opportunities for vaccination (MoVs). The topic is timely and relevant, especially given global efforts under Immunization Agenda 2030 to address stagnating coverage and zero-dose children. The paper is well-structured and methodologically grounded, but certain aspects of data interpretation and framing need refinement to strengthen the study’s contribution to global public health literature.

2. Major Comments

Study Rationale and Framing: The introduction appropriately identifies declining vaccination coverage in Ghana but could more clearly articulate why Tamale Metropolis warrants focused investigation beyond general low coverage in northern Ghana. Adding context on sub-regional inequities or service-delivery characteristics would justify the site selection.

The objectives are stated clearly, but the novelty could be enhanced by framing the study as part of broader post-COVID-19 immunisation recovery or IA2030 monitoring.

Methods:

- The authors mentioned that “multi-stage sampling” was used. This is not enough and should be detailed. What were the stages of sampling they followed, and how did they control against the possibility of introducing biases?

- The use of WHO’s VCQI tool in Stata 15.1 is appropriate and well-documented. However, the denominator definition and handling of missing vaccination records require brief clarification (e.g., whether “not seen” cards were excluded or coded as missing).

- The description of timeliness and invalid doses is clear, but the justification for the <28-day threshold should explicitly cite the WHO reference manual (2018). The authors note that birth doses were excluded; this should be acknowledged as a limitation in interpreting “fully vaccinated” status.

- Please indicate how you handled missingness/missing data. Although you referenced the VCQI tool, none of your tables include information on missing data. If you excluded vaccination data from birth and captured only from 12 to 23 months, that cannot compensate for handling missing data that may have occurred within the age range you captured.

Results: The Tables are clear and consistent with WHO coverage reporting standards. However:

Please add 95% confidence intervals for fully vaccinated/not fully vaccinated in Table 2 to show uncertainty.

Figure 1 should include absolute denominators (n/N) for transparency.

The authors’ reporting of invalid doses and missed opportunities is valuable, but MoV denominators (number of eligible visits) should be clearly defined for reproducibility.

Discussion and Interpretation: The discussion correctly identifies systemic factors affecting coverage (provider practices, caregiver engagement, logistics). However:

Several interpretations (e.g., “female children have lower dropout risk”) cite general Ghanaian evidence that may not derive from the study’s dataset—this should be qualified as secondary literature, not inference. The discussion could benefit from connecting results to specific IA2030 strategic priorities (equity, life-course vaccination, data use). The implications section would be stronger if it linked the findings to concrete local actions (e.g., the use of digital registries, outreach campaigns, and community health volunteers).

Limitations: The limitations section is transparent but can be condensed. Emphasize that reliance on home-based records minimizes recall bias yet limits representativeness for cardless children, potentially underestimating missed opportunities.

Minor Comments:

- Define abbreviations upon their first use, and avoid abbreviations in the Abstract, e.g., MoV, MR1

- State the IA2030 goals/targets in lines 90-91, so that readers can have a better idea of how they compare with the current rates in the population. It is not enough to only cite the document.

- Ensure consistent use of capitalization in vaccine names (e.g., “Penta3,” “OPV3,” “Rota3”).

- Correct minor typographical errors (e.g., repeated “where children where children”.

- The term “not fully vaccinated” might be better described as “partially vaccinated” for readability.

- Verify numeric consistency: weighted sample sizes in Table 2 appear truncated. Please clarify whether it is “111” or “1,111”.

- The reference list is thorough and current (up to 2025), but please ensure journal titles follow PLOS formatting (italicized, full names).

Ethical and Data Transparency: The manuscript states approval from the Navrongo Health Research Centre IRB (NHRCIRB495) and that data are available upon request. Although this is adequate per PLOS Global Public Health standards, the authors should consider making anonymized data openly available via an institutional repository to align with PLOS’s “minimal dataset” policy.

Recommendation: Minor Revision

Rationale: The study is methodologically sound and contributes valuable, recent data on immunisation performance in Ghana. It requires only moderate revision to strengthen contextual framing, clarify methodological details (denominators, exclusions), and refine interpretation to align with IA2030 priorities.

**Do you want your identity to be public for this peer review?** For information about this choice, including consent withdrawal, please see our Privacy Policy..

Reviewer #1: **Yes:**Paul R De Lay, MD, DTM&H (Lond)Paul R De Lay, MD, DTM&H (Lond)Paul R De Lay, MD, DTM&H (Lond)Paul R De Lay, MD, DTM&H (Lond)

Reviewer #2: No

Reviewer #3: **Yes:**Tamar ChitashviliTamar ChitashviliTamar ChitashviliTamar Chitashvili

Reviewer #4: No

Reviewer #5: No

---

## [Editor Report · Decision Letter 1]

19 Feb 2026

PGPH-D-25-02884R1

Routine Childhood Immunization Coverage, Timeliness, Dropout, and Missed Opportunities in Northern Ghana: Evidence from a Community-Based Survey in Tamale Metropolis, Ghana

Dear Dr. Konlan,

Thank you for submitting your manuscript to PLOS Global Public Health.

In your responses to reviewers, you mention things like "this has been addressed." As an editor, that makes it difficult for me to review your manuscript efficiently. Can you re-upload your response to reviewer document and document what specifically you have changed?

We look forward to receiving your revised manuscript.

Kind regards,

Abram L. Wagner, PhD, MPH

Academic Editor
---

## [Editor Report · Decision Letter 2]

18 Mar 2026

Routine Childhood Immunization Coverage, Timeliness, Dropout, and Missed Opportunities in Northern Ghana: Evidence from a Community-Based Survey in Tamale Metropolis, Ghana

PGPH-D-25-02884R2

Dear Mr. Konlan,

We are pleased to inform you that your manuscript 'Routine Childhood Immunization Coverage, Timeliness, Dropout, and Missed Opportunities in Northern Ghana: Evidence from a Community-Based Survey in Tamale Metropolis, Ghana' has been provisionally accepted for publication in PLOS Global Public Health.

Thank you for your patience.

Best regards,

Abram L. Wagner, PhD, MPH

Academic Editor

Reviewer Comments (if any, and for reference):

When you do the final proofing, my recommendation would be to limit certain acronyms, like ZD and MOV, but ultimately that is up to you and the copy editor.